# Asynchronous Stochastic Optimization Robust to Arbitrary Delays

**Alon Cohen**
Tel Aviv University
and Google Research Israel
alonco@tauex.tau.ac.il

**Amit Daniely**
Hebrew University of Jerusalem
and Google Research Israel
amit.daniely@mail.huji.ac.il

**Yoel Drori**
Google Research Israel
dyoel@google.com

**Tomer Koren**
Tel Aviv University
and Google Research Israel
tkoren@tauex.tau.ac.il

**Mariano Schain**
Google Research Israel
marianos@google.com

## Abstract

We consider stochastic optimization with delayed gradients where, at each time step $t$, the algorithm makes an update using a stale stochastic gradient from step $t - d_t$ for some arbitrary delay $d_t$. This setting abstracts asynchronous distributed optimization where a central server receives gradient updates computed by worker machines. These machines can experience computation and communication loads that might vary significantly over time. In the general non-convex smooth optimization setting, we give a simple and efficient algorithm that requires $O(\sigma^2/\epsilon^4 + \tau/\epsilon^2)$ steps for finding an $\epsilon$-stationary point $x$, where $\tau$ is the *average* delay $\frac{1}{T} \sum_{t=1}^{T} d_t$ and $\sigma^2$ is the variance of the stochastic gradients. This improves over previous work, which showed that stochastic gradient decent achieves the same rate but with respect to the *maximal* delay $\max_t d_t$, that can be significantly larger than the average delay especially in heterogeneous distributed systems. Our experiments demonstrate the efficacy and robustness of our algorithm in cases where the delay distribution is skewed or heavy-tailed.

## 1 Introduction

Gradient-based iterative optimization methods are widely used in large-scale machine learning applications as they are extremely simple to implement and use, and come with mild computational requirements. On the other hand, in their standard formulation they are also inherently serial and synchronous due to their iterative nature. For example, in stochastic gradient descent (SGD), each step involves an update of the form $x_{t+1} = x_t - \eta g_t$ where $x_t$ is the current iterate, and $g_t$ is a (stochastic) gradient vector evaluated at $x_t$. To progress to the next step of the method, the subsequent iterate $x_{t+1}$ has to be fully determined by the end of step $t$ as it is required for future gradient queries. Evidently, this scheme has to wait for the computation of the gradient $g_t$ to complete (this is often the most computationally intensive part in SGD) before it can evaluate $x_{t+1}$.

In modern large scale machine learning applications, a direct serial implementation of gradient methods like SGD is overly costly, and parallelizing the optimization process over several cores or machines is desired. Perhaps the most common parallelization approach is via *mini-batching*, where computation of stochastic gradients is distributed across several worker machines that send updates to a parameter server. The parameter server is responsible for accruing the individual updates into a single averaged gradient, and consequently, updating the optimization parameters using this gradient.

While mini-batching is well understood theoretically [e.g., 16, 9, 8, 10], it is still fundamentally synchronous in nature and its performance is adversely determined by the slowest worker machine: the parameter server must wait for all updates from all workers to arrive before it can update the model it maintains. This could cause serious performance issues in heterogeneous distributed networks, where worker machines may be subject to unpredictable loads that vary significantly between workers (due to different hardware, communication bandwidth, etc.) and over time (due to varying users load, power outages, etc.).

An alternative approach that has recently gained popularity is to employ *asynchronous* gradient updates [e.g., 21, 2, 7, 18, 11]; namely, each worker machine computes gradients independently of the other machines, possibly on different iterates, and sends updates to the parameter server in an asynchronous fashion. This implies the parameter server might be making *stale updates* based on *delayed* gradients taken at earlier, out-of-date iterates. While these methods often work well in practice, they have proven to be much more intricate and challenging to analyze theoretically than synchronous gradient methods, and overall our understanding of asynchronous updates remains lacking.

Recently, Arjevani et al. [4] and subsequently Stich and Karimireddy [26] have made significant progress in analyzing delayed asynchronous gradient methods. They have shown that in stochastic optimization, delays only affect a lower-order term in the convergence bounds. In other words, if the delays are not too large, the convergence rate of SGD may not be affected by the delays. (4 first proved this for quadratic objectives; 26 then proved a more general result for smooth functions.) More concretely, Stich and Karimireddy [26] showed that SGD with a sufficiently attenuated step size to account for the delays attains an iteration complexity bound of the form

$$O\left(\frac{\sigma^2}{\epsilon^4} + \frac{\tau_{\max}}{\epsilon^2}\right) \tag{1}$$

for finding an $\epsilon$-stationary point of a possibly non-convex smooth objective function (namely, a point at which the gradient is of norm $\leq \epsilon$). Here $\sigma^2$ is the variance of the noise in the stochastic gradients, and $\tau_{\max}$ is the *maximal* possible delay, which is also needed to be known a-priori for properly tuning the SGD step size. Up to the $\tau_{\max}$ factor in the second term, this bound is identical to standard iteration bounds for stochastic non-convex SGD without delays [12].

While the bound in Eq. (1) is a significant improvement over previous art, it is still lacking in one important aspect: the dependence on the maximal delay could be excessively large in truly asynchronous environments, making the second term in the bound the dominant term. For example, in heterogeneous or massively distributed networks, the maximal delay is effectively determined by the single slowest (or less reliable) worker machine—which is precisely the issue with synchronous methods we set to address in the first place. Moreover, as Stich and Karimireddy [26] show, the step size used to achieve the bound in Eq. (1) could be as much as $\tau_{\max}$-times smaller than that of without delays, which could severely impact performance in practice.

## 1.1 Contribution

We propose a new algorithm for stochastic optimization with asynchronous delayed updates, we call "Picky SGD," that is significantly more robust than SGD, especially when the (empirical) distribution of delays is skewed or heavy-tailed and thus the maximal delay could be very large. For general smooth possibly non-convex objectives, our algorithm achieves a convergence bound of the form

$$O\left(\frac{\sigma^2}{\epsilon^4} + \frac{\tau_{\text{avg}}}{\epsilon^2}\right),$$

where now $\tau_{\text{avg}}$ is the *average* delay in retrospect. This is a significant improvement over the bound in Eq. (1) whenever $\tau_{\text{avg}} \ll \tau_{\max}$, which is indeed the case with heavy-tailed delay distributions. Moreover, Picky SGD is very efficient, extremely simple to implement, and does *not* require to know the average delay $\tau_{\text{avg}}$ ahead of time for optimal tuning. In fact, the algorithm only relies on a single additional hyper-parameter beyond the step-size.

Notably, and in contrast to SGD as analyzed in previous work [26], our algorithm is able to employ a significantly larger effective step size, and thus one could expect it to perform well in practice compared to SGD. Indeed, we show in experiments that Picky SGD is able to converge quickly on large image classification tasks with a relatively high learning rate, even when very large delays are

introduced. In contrast, in the same setting, SGD needs to be configured with a substantially reduced step size to be able to converge at all, consequently performing poorly compared to our algorithm.

Finally, we also address the case where $f$ is smooth and convex, in which we give a close variant of our algorithm with an iteration complexity bound of the form

$$O\left(\frac{\sigma^2}{\epsilon^2} + \frac{\tau_{\text{avg}}}{\epsilon}\right)$$

for obtaining a point $x$ with $f(x) - f(x^*) \leq \epsilon$ (where $x^*$ is a minimizer of $f$ over $\mathbb{R}^d$). Here as well, our rate matches precisely the one obtained by the state-of-the-art [26], but with the dependence on the maximal delay being replaced with the average delay. For consistency of presentation, we defer details on the convex case to the full version of the paper [?] and focus here on our algorithm for non-convex optimization.

Concurrently to this work, Aviv et al. [5] derived similar bounds that depend on the average delay. Compared to our contribution, their results are adaptive to the smoothness and noise parameters, but on the other hand, are restricted to convex functions and their algorithms are more elaborate and their implementation is more involved.

## 1.2 Additional related work

For general background on distributed asynchronous optimization and basic asymptotic convergence results, we refer to the classic book by Bertsekas and Tsitsiklis [6]. Since the influential work of Niu et al. [24], there has been significant interest in asynchronous algorithms in a related model where there is a delay in updating individual *parameters* in a shared parameter vector (e.g., [25, 19, 28, 17]). This is of course very different from our model, where steps use the full gradient vector in atomic, yet delayed, updates.

Also related to our study is the literature on Local SGD (e.g., 27 and references therein), which is a distributed gradient method that perform several local (serial) gradient update steps before communicating with the parameter server or with other machines. Local SGD methods have become popular recently since they are used extensively in Federated Learning [20]. We note that the theoretical study in this line of work is mostly concerned with analyzing *existing* distributed variants of SGD used in practice, whereas we aim to develop and analyze *new* algorithmic tools to help with mitigating the effect of stale gradients in asynchronous optimization.

A related yet orthogonal issue in distribution optimization, which we do not address here, is reducing the communication load between the workers and servers. One approach that was recently studied extensively is doing this by compressing gradient updates before they are transmitted over the network. We refer to [3, 14, 26] for further discussion and references.

## 2 Setup and Basic Definitions

### 2.1 Stochastic non-convex smooth optimization

We consider stochastic optimization of a $\beta$-smooth (not necessarily convex) non-negative function $f$ defined over the $d$-dimensional Euclidean space $\mathbb{R}^d$. A function $f$ is said to be $\beta$-smooth if it is differentiable and its gradient operator is $\beta$-Lipschitz, that is, if $\|\nabla f(x) - \nabla f(y)\| \leq \beta\|x - y\|$ for all $x, y \in \mathbb{R}^d$. This in particular implies (e.g., [22]) that for all $x, y \in \mathbb{R}^d$,

$$f(y) \leq f(x) + \nabla f(x) \cdot (y - x) + \frac{\beta}{2}\|y - x\|^2. \tag{2}$$

We assume a stochastic first-order oracle access to $f$; namely, $f$ is endowed with a stochastic gradient oracle that given a point $x \in \mathbb{R}^d$ returns a random vector $\tilde{g}(x)$, independent of all past randomization, such that $\mathbb{E}[\tilde{g}(x) \mid x] = \nabla f(x)$ and $\mathbb{E}[\|\tilde{g}(x) - \nabla f(x)\|^2 \mid x] \leq \sigma^2$ for some variance bound $\sigma^2 \geq 0$. In this setting, our goal is to find an $\epsilon$-stationary point of $f$, namely, a point $x \in \mathbb{R}^d$ such that $\|\nabla f(x)\| \leq \epsilon$, with as few samples of stochastic gradients as possible.

### 2.2 Asynchronous delay model

We consider an abstract setting where stochastic gradients (namely, outputs for invocations of the stochastic first-order oracle) are received asynchronously and are subject to arbitrary delays. The asynchronous model can be abstracted as follows. We assume that at each step $t$ of the optimization,

the algorithm obtains a pair $(x_{t-d_t}, g_t)$ where $g_t$ is a stochastic gradient at $x_{t-d_t}$ with variance bounded by $\sigma^2$; namely, $g_t$ is a random vector such that $\mathbb{E}_t g_t = \nabla f(x_{t-d_t})$ and $\mathbb{E}_t \|g_t - \nabla f(x_{t-d_t})\|^2 \leq \sigma^2$ for some delay $0 \leq d_t < t$. Here and throughout, $\mathbb{E}_t[\cdot]$ denotes the expectation conditioned on all randomness drawn before step $t$. After processing the received gradient update, the algorithm may query a new stochastic gradient at whatever point it chooses (the result of this query will be received with a delay, as above).

Few remarks are in order:

- We stress that the delays $d_1, d_2, \ldots$ are entirely arbitrary, possibly chosen by an adversary; in particular, we do *not* assume they are sampled from a fixed stationary distribution. Nevertheless, we assume that the delays are independent of the randomness of the stochastic gradients (and of the internal randomness of the optimization algorithm, if any).[1]
- For simplicity, we assumed above that a stochastic gradient is received at every round $t$. This is almost without loss of generality:[2] if at some round no feedback is observed, we may simply skip the round without affecting the rest of the optimization process (up to a re-indexing of the remaining rounds).
- Similarly, we will also assume that only a single gradient is obtained in each step; the scenario that multiple gradients arrive at the same step (as in mini-batched methods) can be simulated by several subsequent iterations in each of which a single gradient is processed.

## 3   The Picky SGD Algorithm

We are now ready to present our asynchronous stochastic optimization algorithm, which we call Picky SGD; see pseudo-code in Algorithm 1. The algorithm is essentially a variant of stochastic gradient descent, parameterized by a learning rate $\eta$ as well as a target accuracy $\epsilon$.

---

**Algorithm 1:** Picky SGD

1: **input**: learning rate $\eta$, target accuracy $\epsilon$.
2: **for** $t = 1, \ldots, T$ **do**
3:     **receive** delayed stochastic gradient $g_t$ and point $x_{t-d_t}$ such that $\mathbb{E}_t[g_t] = \nabla f(x_{t-d_t})$.
4:     **if** $\|x_t - x_{t-d_t}\| \leq \epsilon/(2\beta)$ **then**
5:         **update:** $x_{t+1} = x_t - \eta g_t$.
6:     **else**
7:         **pass:** $x_{t+1} = x_t$.
8:     **end if**
9: **end for**

---

Picky SGD maintains a sequence of iterates $x_1, \ldots, x_T$. At step $t$, the algorithm receives a delayed stochastic gradient $g_t$ that was computed at an earlier iterate $x_{t-d_t}$ (line 3). Then, in line 4, the algorithm tests whether $\|x_t - x_{t-d_t}\| \leq \epsilon/2\beta$. Intuitively, this aims to verify whether the delayed (expected) gradient $\nabla f(x_{t-d_t})$ is "similar" to the gradient $\nabla f(x_t)$ at the current iterate $x_t$; due to the smoothness of $f$, we expect that if $x_{t-d_t}$ is close to $x_t$, then also the corresponding gradients will be similar. If this condition holds true, the algorithm takes a gradient step using $g_t$ with step size $\eta$.

Our main theoretical result is the following guarantee on the success of the algorithm.

**Theorem 1.** *Suppose that Algorithm 1 is initialized at $x_1 \in \mathbb{R}^d$ with $f(x_1) \leq F$ and ran with*

$$T \geq 500 \beta F \left( \frac{\sigma^2}{\epsilon^4} + \frac{\tau + 1}{\epsilon^2} \right), \quad \eta = \frac{1}{4\beta} \min\left\{ 1, \frac{\epsilon^2}{\sigma^2} \right\},$$

*where $\tau$ be the average delay, i.e., $\tau = (1/T) \sum_{t=1}^{T} d_t$. Then, with probability at least $\frac{1}{2}$, there is some $1 \leq t \leq T$ for which $\|\nabla f(x_t)\| \leq \epsilon$.*

Observe that the optimal step size in Theorem 1 is independent of the average delay $\tau$. This is important for two main reasons: (i) implementing the algorithm does not require knowledge about

---

[1] One can thus think of the sequence of delays as being fixed ahead of time by an oblivious adversary.

[2] We may, in principle, allow to query the stochastic gradient oracle even on rounds where no feedback is received, however this would be redundant in most reasonable instantiations of this model (e.g., in a parameter server architecture).

future, yet-to-be-seen delays; and (ii) even with very large delays, the algorithm can maintain a high effective step size.

We note that the guarantee of Theorem 1 is slightly different from typical bounds in non-convex optimization (e.g., the bounds appearing in the previous work [14]): our result claims about the *minimal* gradient norm of any iterate rather than the *average* gradient norm over the iterates. Arguably, this difference does not represent a very strong limitation: the significance of convergence bounds in non-convex optimization is, in fact, in that they ensure that one of the iterates along the trajectory of the algorithm is indeed an approximate critical point, and the type of bound we establish is indeed sufficient to ensure exactly that.

We further note that while the theorem above only guarantees a constant success probability, it is not hard to amplify this probability to an arbitrary $1 - \delta$ simply by restarting the algorithm $O(\log(1/\delta))$ times (with independent stochastic gradients); with high probability, one of the repetitions will be successful and run through a point with gradient norm $\leq \epsilon$, which would imply the guarantee in the theorem with probability at least $1 - \delta$.

## 4 Analysis

In this section we analyze Algorithm 1 and prove our main result. Throughout, we denote $x'_t = x_{t-d_t}$ and let $N_t$ denote the noise vector at step $t$, namely $N_t = g_t - \nabla f(x'_t)$. Note that $\mathbb{E}[N_t \mid x_t, x'_t] = 0$ and $\mathbb{E}[\|N_t\|^2 \mid x_t, x'_t] \leq \sigma^2$, since the iterates $x_t, x'_t$ are conditionally independent of the noise in $g_t$ as this gradient is obtained by the algorithm only at step $t$, after $x_t, x'_t$ were determined.

To prove Theorem 1, we will analyze a variant of the algorithm that will stop making updates once it finds a point with $\|\nabla f(x)\| \leq \epsilon$ (and eventually *fails* otherwise). That is, if $\|x_t - x'_t\| > \epsilon/2\beta$ or $\|\nabla f(x_t)\| \leq \epsilon$ then $x_{t+1} = x_t$. Else, $x_{t+1} = x_t - \eta g_t$. This variant is impossible to implement (since it needs to compute the exact gradient at each step), but the guarantee of Theorem 1 is valid for this variant if and only if it is valid for the original algorithm: one encounters an $\epsilon$-stationary point if and only if the other does so.

First, we prove a simple technical lemma guaranteeing that whenever the algorithm takes a step, a large gradient norm implies a large decrease in function value. It is a variant of the classical "descent lemma," adapted to the case where the gradient step is taken with respect to a gradient computed at a nearby point.

**Lemma 2.** *Fix $x, x' \in \mathbb{R}^d$ with $\|x - x'\| \leq \epsilon/2\beta$ and $\|\nabla f(x')\| > \epsilon$. Let $N \in \mathbb{R}^d$ be a random vector with $\mathbb{E}[N \mid x, x'] = 0$ and $\mathbb{E}[\|N\|^2 \mid x, x'] \leq \sigma^2$. Then,*

$$\mathbb{E}[f(x - \eta(\nabla f(x') + N))] - \mathbb{E}f(x) \leq -\frac{\eta}{2}\mathbb{E}\|\nabla f(x')\|^2 + \frac{\eta^2\beta}{2}(\sigma^2 + \mathbb{E}\|\nabla f(x')\|^2).$$

*In particular, for our choice of $\eta$, we have*

$$\frac{\eta}{4}\mathbb{E}\|\nabla f(x')\|^2 \leq \mathbb{E}f(x) - \mathbb{E}[f(x - \eta(\nabla f(x') + N))]. \tag{3}$$

*Proof.* Using the smoothness of $f$ (Eq. (2)), we have

$$f(x - \eta(\nabla f(x') + N)) - f(x) \leq -\eta\nabla f(x) \cdot (\nabla f(x') + N) + \tfrac{1}{2}\eta^2\beta\|\nabla f(x') + N\|^2.$$

Taking expectation over $N$ conditioned on $x, x'$, we get

$$\begin{aligned}
&\mathbb{E}[f(x - \eta(\nabla f(x') + N)) - f(x) \mid x, x'] \\
&\quad \leq -\eta\nabla f(x) \cdot \nabla f(x') + \tfrac{1}{2}\eta^2\beta(\|\nabla f(x')\|^2 + \sigma^2) \\
&\quad = -\eta\nabla f(x') \cdot \nabla f(x') - \eta\nabla f(x') \cdot (\nabla f(x) - \nabla f(x')) + \tfrac{1}{2}\eta^2\beta(\|\nabla f(x')\|^2 + \sigma^2) \\
&\quad \leq -\eta\|\nabla f(x')\|^2 + \eta\beta\|\nabla f(x')\|\|x - x'\| + \tfrac{1}{2}\eta^2\beta(\|\nabla f(x')\|^2 + \sigma^2) \\
&\quad = \eta(\beta\|\nabla f(x')\|\|x - x'\| - \|\nabla f(x')\|^2) + \tfrac{1}{2}\eta^2\beta(\|\nabla f(x')\|^2 + \sigma^2).
\end{aligned}$$

Since $\epsilon \leq \|\nabla f(x')\|$ then

$$\|x - x'\| \leq \frac{\epsilon}{2\beta} \leq \frac{1}{2\beta}\|\nabla f(x')\|,$$

and we have

$$\mathbb{E}\big[f(x - \eta(\nabla f(x') + N)) - f(x) \mid x, x'\big] \le -\frac{\eta}{2}\|\nabla f(x')\|^2 + \tfrac{1}{2}\eta^2\beta(\sigma^2 + \|\nabla f(x')\|^2).$$

If $\epsilon \ge \sigma$ then $\sigma^2 \le \|\nabla f(x')\|^2$. This, with $\eta = 1/4\beta$, yields Eq. (3). If $\epsilon < \sigma$ and $\eta = \epsilon^2/4\sigma^2\beta$, then $\eta^2 \le \epsilon^2/16\sigma^2\beta^2$. Plugging that in instead, using $\|\nabla f(x')\| \ge \epsilon$, and taking expectations (with respect to $x, x'$) gets us Eq. (3). $\blacksquare$

We next introduce a bit of additional notation. We denote by $I_t$ the indicator of event that the algorithm performed an update at time $t$. Namely,

$$I_t = I\big\{\|x_t - x'_t\| \le \epsilon/2\beta \text{ and } \|\nabla f(x_t)\| > \epsilon\big\}.$$

Note that $I_t = 1$ implies that $\|\nabla f(x_s)\| \ge \epsilon$ for all $s = 1, \dots, t$. Further, we denote by $\Delta_t = f(x_t) - f(x_{t+1})$ the improvement at time $t$. Since $f$ is non-negative and $f(x_1) \le F$, we have that for all $t$,

$$\sum_{i=1}^{t} \Delta_i = f(x_1) - f(x_{t+1}) \le F.$$

Note that by Lemma 2 we have that $\mathbb{E}\Delta_t \ge 0$. The rest of the proof is split into two cases: $\sigma \le \epsilon$, and $\sigma \ge \epsilon$.

## 4.1 Case (i): $\sigma \le \epsilon$

This regime is intuitively the "low noise" regime in which the standard deviation of the gradient noise, $\sigma$, is smaller than the desired accuracy $\epsilon$. We prove the following.

**Lemma 3.** *Suppose that $\sigma \le \epsilon$ and the algorithm fails with probability $\ge \frac{1}{2}$. Then $T \le 128\beta F(\tau + 1)/\epsilon^2$.*

To prove the lemma above, we first show that the algorithm must make a significant number of updates, as shown by the following lemma.

**Lemma 4.** *If the algorithm fails, then the number of updates that it makes is at least $T/4(\tau + 1)$.*

*Proof.* Consider $U_{2\tau}$, the number of steps $t$ for which the delay $d_t$ is at least $2\tau$. We must have $U_{2\tau} \le T/2$ (otherwise the total sum of delays exceeds $\tau T$, contradicting the definition of $\tau$). On the other hand, let $k$ be the number of updates that the algorithm makes. Let $t_1 < t_2 < \dots < t_k$ be the steps in which an update is made. Denote $t_0 = 0$ and $t_{k+1} = T$. Now, fix $i$ and consider the steps at times $s_n = t_i + n$ for $n \in [1, 2, \dots, t_{i+1} - t_i - 1]$. In all those steps no update takes place and $x_{s_n} = x_{t_i}$. We must have $d_{s_n} > n$ for all $n$ (otherwise $x_t = x_{t-d_t}$ for $t = s_n$ and an update occurs). In particular we have that $d_{s_n} \ge 2\tau$ in at least $t_{i+1} - t_i - 1 - 2\tau$ steps in $[t_i, t_{i+1}]$. Hence,

$$U_{2\tau} \ge \sum_{i=0}^{k-1} (t_{i+1} - t_i - 1 - 2\tau) = T - k(1 + 2\tau).$$

Finally, it follows that $T - k(1 + 2\tau) \le T/2$ which implies $k \ge \frac{T}{4(\tau+1)}$. $\blacksquare$

Given the lemma above, we prove Lemma 3 by showing that if the algorithm fails, it makes many updates in all of which we have $\|\nabla f(x_t)\| > \epsilon$. By Lemma 2, this means that in the $T$ time steps of the algorithm, it must decrease the value of $f$ significantly. Since we start at a point in which $f(x_1) \le F$, we must conclude that $T$ cannot be too large.

*Proof of Lemma 3.* Combining Eq. (3) with $\eta = 1/(4\beta)$ and Lemma 4, we get that if the algorithm fails with probability $\ge \frac{1}{2}$ then

$$F \ge \sum_{t=1}^{T} \mathbb{E}\Delta_t \ge \frac{1}{16\beta} \sum_{t=1}^{T} \mathbb{E}[I_t\|\nabla f(x_t)\|^2] \ge \frac{1}{16\beta}\mathbb{E}\left[\sum_{t=1}^{T} I_t\|\nabla f(x_t)\|^2\right]$$

$$\ge \frac{1}{32\beta}\mathbb{E}\left[\sum_{t=1}^{T} I_t\|\nabla f(x_t)\|^2 \;\middle|\; \text{algorithm fails}\right] \ge \frac{\epsilon^2}{32\beta}\mathbb{E}\left[\sum_{t=1}^{T} I_t \;\middle|\; \text{algorithm fails}\right] \ge \frac{\epsilon^2}{32\beta}\frac{T}{4(\tau+1)}.$$

This yields the lemma's statement. $\blacksquare$

## 4.2 Case (ii): $\sigma > \epsilon$

This is the "high noise" regime. For this case, we prove the following guarantee for the convergence of our algorithm.

**Lemma 5.** *Assume that $\sigma > \epsilon$ and the algorithm fails with probability $\geq \frac{1}{2}$. Then,*

$$\sum_{t=1}^{T} \mathbb{E}\Delta_t \geq \frac{T}{500\beta} \min\left\{\frac{\epsilon^2}{\tau}, \frac{\epsilon^4}{\sigma^2}\right\}.$$

*In particular,*

$$T \leq 500\beta F\left(\frac{\tau}{\epsilon^2} + \frac{\sigma^2}{\epsilon^4}\right).$$

This result is attained using the following observation. Consider the iterate of algorithm at time $t$, $x_t$, and the point at which the gradient was computed $x_t' = x_{t-d_t}$. We claim that if the algorithm has not decreased the function value sufficiently during the interval $[t - d_t, t - 1]$, then it is likely to trigger a large decline in the function value at time $t$. Formally, either $\mathbb{E}\Delta_t$ is large, or $\sum_{i=t-d_t}^{t-1} \mathbb{E}\Delta_i$ is large. To show the claim, we first upper bound the distance $\|x_t - x_t'\|$ in terms of $\sum_{i=t-d_t}^{t-1} \mathbb{E}\Delta_i$, as shown by the following technical lemma.

**Lemma 6.** *For all $t$ and $k$, it holds that*

$$\mathbb{E}\|x_t - x_{t+k}\| \leq \sqrt{\frac{1}{\beta} \sum_{i=t}^{t+k-1} \mathbb{E}\Delta_i} + \frac{4}{\epsilon} \sum_{i=t}^{t+k-1} \mathbb{E}\Delta_i.$$

*Proof.* We have

$$\mathbb{E}\|x_t - x_{t+k}\| = \eta\mathbb{E}\left\|\sum_{i=t}^{t+k-1} I_i(\nabla f(x_i') + N_i)\right\| \leq \eta\mathbb{E}\left\|\sum_{i=t}^{t+k-1} I_i\nabla f(x_i')\right\| + \eta\mathbb{E}\left\|\sum_{i=t}^{t+k-1} I_i N_i\right\|.$$

We continue bounding the second term above as follows:

$$
\begin{aligned}
\mathbb{E}\left\|\sum_{i=t}^{t+k-1} I_i N_i\right\| &\leq \sqrt{\mathbb{E}\left\|\sum_{i=t}^{t+k-1} I_i N_i\right\|^2} \\
&= \sqrt{\mathbb{E}\sum_{i=t}^{t+k-1}\sum_{j=t}^{t+k-1} I_i I_j N_i \cdot N_j} \\
&= \sqrt{\mathbb{E}\sum_{i=t}^{t+k-1} I_i\|N_i\|^2} &&(\mathbb{E}[N_i \mid I_i, I_j, N_j] = 0 \text{ for } i > j) \\
&\leq \sigma\sqrt{\mathbb{E}\sum_{i=t}^{t+k-1} I_i} \\
&\leq \frac{\sigma}{\epsilon}\sqrt{\mathbb{E}\sum_{i=t}^{t+k-1} I_i\|\nabla f(x_i')\|^2} &&(\|\nabla f(x_i')\| \geq \epsilon \text{ when } I_i = 1) \\
&\leq \frac{\sigma}{\epsilon}\sqrt{\frac{16\sigma^2\beta}{\epsilon^2} \sum_{i=t}^{t+k-1} \mathbb{E}\Delta_i} &&(\text{Eq. (3)}, \eta = \epsilon^2/4\beta\sigma^2) \\
&= \frac{4\sigma^2}{\epsilon^2}\sqrt{\beta \sum_{i=t}^{t+k-1} \mathbb{E}\Delta_i}
\end{aligned}
$$

$$= \frac{1}{\eta} \sqrt{\frac{1}{\beta} \sum_{i=t}^{t+k-1} \mathbb{E}\Delta_i}, \qquad\qquad (\eta = \epsilon^2/4\beta\sigma^2)$$

and

$$\mathbb{E} \left\| \sum_{i=t}^{t+k-1} I_i \nabla f(x_i') \right\| \leq \sum_{i=t}^{t+k-1} \mathbb{E}I_i \|\nabla f(x_i')\|$$

$$\leq \frac{1}{\epsilon} \sum_{i=t}^{t+k-1} \mathbb{E}I_i \|\nabla f(x_i')\|^2 \qquad (\|\nabla f(x_i')\| \geq \epsilon \text{ when } I_i = 1)$$

$$\leq \frac{4}{\epsilon\eta} \sum_{i=t}^{t+k-1} \mathbb{E}\Delta_i. \qquad\qquad\qquad (\text{Eq. (3)})$$

This completes the proof. ∎

Given the lemma above, it is now clear that if $\sum_{i=t-d_t}^{t-1} \mathbb{E}\Delta_i$ is sufficiently small, then $\mathbb{E}\|x_t - x_t'\| \ll \epsilon/\beta$ which means that the algorithm is likely (with constant probability) to take a step at time $t$. This argument yields the following.

**Corollary 7.** *Assume that the algorithm fails with probability $\geq \frac{1}{2}$. If $\sum_{i=t-d_t}^{t-1} \mathbb{E}\Delta_i < \epsilon^2/125\beta$ then $\mathbb{E}\Delta_t \geq \epsilon^4/64\sigma^2\beta$. In particular,*

$$\mathbb{E}\Delta_t + \frac{1}{2\tau} \sum_{i=t-d_t}^{t-1} \mathbb{E}\Delta_i \geq \frac{1}{250\beta} \min\left\{ \frac{\epsilon^2}{\tau}, \frac{\epsilon^4}{\sigma^2} \right\}.$$

*Proof.* If $\sum_{i=t-d_i}^{t-1} \mathbb{E}\Delta_i < \epsilon^2/125\beta$, then $\mathbb{E}\|x_{t-d_t} - x_t\| \leq \epsilon/8\beta$ by Lemma 6. By a Markov inequality, with probability $\geq \frac{3}{4}$, we have $\|x_{t-d_t} - x_t\| \leq \epsilon/2\beta$. Since the probability that $\|\nabla f(x_{t-d_t})\| > \epsilon$ is at least $\frac{1}{2}$, we get that $\mathbb{E}I_t \geq \frac{1}{4}$. By Lemma 2 this implies that

$$\mathbb{E}\Delta_t \geq \frac{1}{4} \cdot \frac{\epsilon^2 \cdot \epsilon^2}{16\sigma^2\beta} = \frac{\epsilon^4}{64\sigma^2\beta},$$

which yields our claim. ∎

We now prove our main claim. We show that if the algorithm fails, then in *all* time steps in which $d_t \leq 2\tau$ (of which there are at least $T/2$), either the algorithm makes a substantial step, or it has made significant updates in the interval $[t - d_t, t - 1]$. In any case, the function value must necessarily decrease overall in the $T$ time steps of the algorithm, concluding that $T$ cannot be too large.

*Proof of Lemma 5.* We have,

$$\sum_{t=1}^{T} \mathbb{E}\Delta_t \geq \sum_{t:d_t \leq 2\tau} \frac{1}{2\tau} \sum_{i=t-d_t}^{t-1} \mathbb{E}\Delta_i.$$

Hence, using Corollary 7,

$$\sum_{t=1}^{T} \mathbb{E}\Delta_t \geq \frac{1}{2} \sum_{t:d_t \leq 2\tau} \left( \mathbb{E}\Delta_t + \frac{1}{2\tau} \sum_{i=t-d_t}^{t-1} \mathbb{E}\Delta_i \right)$$

$$\geq |\{t : d_t \leq 2\tau\}| \frac{1}{250\beta} \min\left\{ \frac{\epsilon^2}{\tau}, \frac{\epsilon^4}{\sigma^2} \right\}$$

$$\geq \frac{T}{2} \frac{1}{250\beta} \min\left\{ \frac{\epsilon^2}{\tau}, \frac{\epsilon^4}{\sigma^2} \right\}$$

$$= \frac{T}{500\beta} \min\left\{ \frac{\epsilon^2}{\tau}, \frac{\epsilon^4}{\sigma^2} \right\},$$

where we used Markov's inequality to show that $|\{t : d_t \leq 2\tau\}| \geq \frac{1}{2}T$. ∎

## 4.3 Concluding the proof

*Proof of Theorem 1.* In the case $\sigma \leq \epsilon$, Lemma 3 implies that if $T > 128\beta F(\tau + 1)/\epsilon^2$ then the algorithms succeeds with probability greater than $1/2$, which yields the theorem in this case. Similarly, Lemma 5 gives our claim in the case when $\sigma > \epsilon$. ∎

## 5 Experiments

To illustrate the robustness and efficacy of Picky SGD, we present a comparison between the performance of SGD versus Picky SGD under various delay distributions. In particular, we show that Picky SGD requires significantly less iterations to reach a fixes goal and is more robust to varying delay distributions.

### 5.1 Setup

The main goal of our experimental setup is to be reproducible. For that end, the experimentation is done in two phases. First, we perform a simulation to determine the delay $d_t$ at each iteration without actually computing any gradients:[3] this is done by simulating $N$ concurrent worker threads sharing and collectively advancing a global iteration number, where each worker repeatedly records the current global iteration number $t_{\text{start}}$, waits a random amount of time from a prescribed Poisson distribution, then records the new global iteration number $t = t_{\text{end}}$ and the difference $d_t = t_{\text{end}} - t_{\text{start}}$, and increases the global iteration number. This information (a *delay schedule*) is calculated once for each tested scheme (differing in the number of workers and random distribution, as detailed below), and is stored for use in the second phase.

In the second phase of the experiments, the algorithms SGD and Picky SGD are executed for each delay schedule. Here, at every iteration the gradient is computed (if needed) and is kept until its usage as dictated by the schedule (and then applied at the appropriate global iteration number). As a result of this configuration, we get a fully reproducible set of experiments, where the algorithms performance may be compared as they are executed over identical delay series of identical statistical properties.

We created four different delay schedules: A baseline schedule (A) using $N = 10$ workers and sampling the simulated wait from a Poisson distribution (this schedule serves to compare Picky SGD and SGD in a setting of relatively small delay variance) and schedules (B) (C) and (D) all using $N = 75$ workers and sampling the simulated wait from bi-modal mixtures of Poisson distributions of similar mean but increasing variance respectively.[4] See Figure 2 in the the full version of the paper [**?** ] for an illustration of the delay distributions of the four delay schedules used.

All training is performed on the standard CIFAR-10 dataset [15] using a ResNet56 with 9 blocks model [13] and implemented in TensorFlow [1]. We compare Picky SGD (Algorithm 1) to the SGD algorithm which unconditionally updates the state $x_t$ given the stochastic delayed gradient $g_t$ (recall that $g_t$ is the stochastic gradient at state $x_{t-d_t}$).

For both algorithms, instead of a constant learning rate $\eta$ we use a piecewise-linear learning rate schedule as follows: we consider a baseline $\eta_0$ piecewise-linear learning rate schedule[5] that achieves optimal performance in a synchronous distributed optimization setting (that is, for $d_t \equiv 0$)[6] and search (for each of the four delay schedules and each algorithm – to compensate for the effects of delays) for the best multiple of the baseline rate and the best first rate-change point. Alternatively, we also used a cosine decay learning rate schedule (with the duration of the decay as meta parameters). Another meta-parameter we optimize is the threshold $\epsilon/(2\beta)$ in line 4 of Picky SGD. Batch size 64 was used throughout the experiments. Note that although use chose the threshold value $\epsilon/2\beta$ by an exhaustive search, in practice, a good choice can be found by logging the distance values during a typical execution and choosing a high percentile value. See the full version of the paper [**?** ] for more details.

---

[3]Note that up to the training data ordering a computation of $T$ steps of Picky SGD or SGD is uniquely determined by the starting state $x_1$ and the sequence $\{t - d_t\}_{t=1...T}$.

[4]See the the full version of the paper [**?** ] for specific parameter values and implementation details.

[5]With rate changes at three achieved accuracy points 0.93, 0.98, and 0.99.

[6]This is also the best performance achievable in an asynchronous setting.

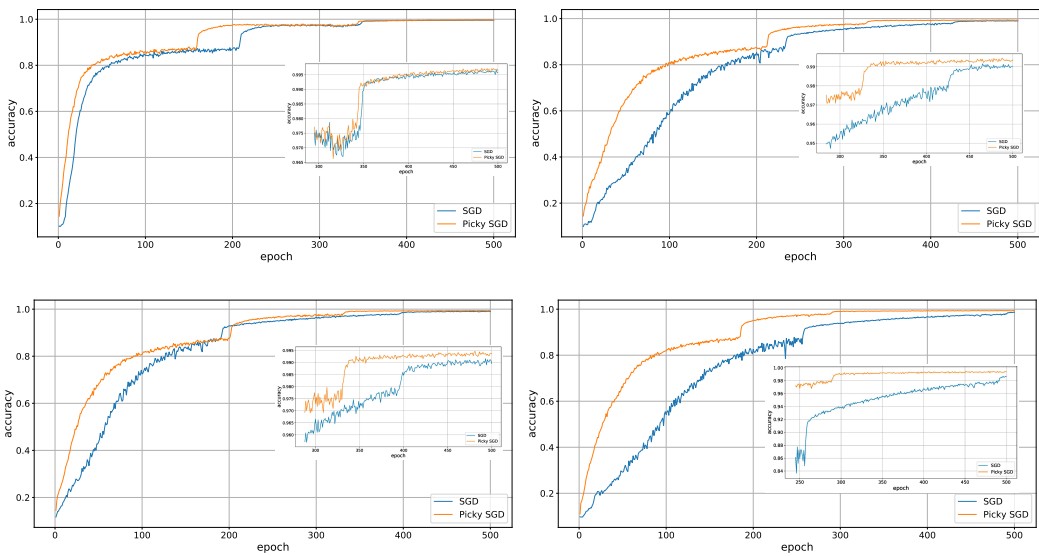

Figure 1: Accuracy trajectory (with a zoom-in on the tail of the convergence) over train epochs for the four delay schedules of Fig. 2, respectively: the key metrics (reported in Table 1) for each trajectory are epochs to reach 0.99 accuracy (the number of epochs required to reach the 0.99 accuracy mark) and the baseline learning rate multiplier $\eta/\eta_0$.

## 5.2 Results

The accuracy trajectory for the best performing combination of parameters of each algorithm for each of the four delay schedules is shown in Fig. 1 and summarized in Table 1. Clearly, Picky SGD significantly outperforms SGD in terms of the final accuracy and the number of epochs it takes to achieve it. We also emphasize that the generalization performance (that is, the evaluation accuracy as related to the training accuracy) was not observed to vary across delay schedules or the applied algorithms (see e.g., Fig. 4 in the the full version of the paper [? ]), and that the nature of the results is even more pronounced when using the alternative cosine decay learning rate schedule (see Fig. 5 in the the full version of the paper [? ]). Specific details of the meta parameters used, and additional performance figures are reported in the full version of the paper [? ].

Table 1: Summary of the key metrics from Fig. 1, for each of the four delay schedules A, B, C, and D .

|   | Epochs to 0.99% | | LR multiplier ($\eta/\eta_0$) | |
|---|---|---|---|---|
|   | Picky SGD | SGD | Picky SGD | SGD |
| A | 344 | 350 | 0.5 | 0.5 |
| B | 333 | 451 | 0.2 | 0.05 |
| C | 337 | 438 | 0.2 | 0.05 |
| D | 288 | 466 | 0.2 | 0.05 |

## 5.3 Discussion

We first observe that while the number of epochs it takes Picky SGD to reach the target accuracy mark is almost the same across the delay schedules (ranging from 288 to 344), SGD requires significantly more epochs to attain the target accuracy (ranging from 350 up to 466 for the highest variance delay schedule)—this is consistent with the average-delay bound dependence of Picky SGD (as stated in Theorem 1) compared to the max-delay bound dependence of SGD. Furthermore, the best baseline learning rate multiplier meta-parameter for Picky SGD is the same (0.2) across all high-variance delay schedules, while the respective meta parameter for SGD is significantly smaller (0.05) and sometimes varying, explaining the need for more steps to reach the target and evidence of Picky SGD superior robustness.

**Acknowledgements**

AD is partially supported by the Israeli Science Foundation (ISF) grant no. 2258/19. TK is partially supported by the Israeli Science Foundation (ISF) grant no. 2549/19, by the Len Blavatnik and the Blavatnik Family foundation, and by the Yandex Initiative in Machine Learning.

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
