# OpenReview forum: "Asynchronous Stochastic Optimization Robust to Arbitrary Delays"
_NeurIPS.cc/2021/Conference — NeurIPS 2021 Poster_

### Official Review · Reviewer_v3Nq · 2021-07-10

**Rating:** 5
**Confidence:** 3

**Summary:**

This paper proposes a distributed SGD algorithm, called Picky SGD, for asynchronous implementation of SGD with multiple workers and arbitrary delays. Compared to prior works which analyzed a plain SGD algorithm, the main advantage of picky SGD is that it has a tighter complexity bound of $O(\sigma^2/\epsilon^4 + \tau_{avg}/\epsilon^2)$ instead of $O(\sigma^2/\epsilon^4 + \tau_{max}/\epsilon^2)$ as in the prior work.

**Limitations And Societal Impact:**

There isn't any potential negative societal impact. The limitation/suggestions for improvement have been discussed in the main review above.

**Main Review:**

As mentioned above, the main contribution of this paper is a new algorithm with improved complexity bound with respect to the delays in the asynchronous algorithm. The main idea is to perform a selective SGD update only made when the transmitted stochastic gradient is computed from an iterate that is "close" to the current one at the server. The proof seems to be follow from standard analysis of distributed SGD (and actually SGD in general). Numerical experiments seem to indicate a better performance of the proposed algorithm compared to standard SGD.

The reviewer has the following comments:

- The reviewer is concerned with the practicality of the proposed algorithm and the theoretical result provided in the study.

First, it should be noted that at every update, the worker has to send both $x_{t-d_t}$ and $g_t$ to the server which involves a doubled communication cost. In addition, given this additional requirement on the communication protocol, it is unclear if the proposed algorithm can be extended to be used with compression, e.g., as studied in [25].

Second, the current proof in the paper only analyzes the time complexity with a success probability of greater than 1/2. After Theorem 1, it is stated that this success probability can be "amplified" to $1-\delta$ for any $\delta > 0$ by repeating the algorithm with $T$ iterations and "running through a point with gradient norm $\leq \epsilon$". Such scheme appears to be impractical since it involves checking the gradient norms of the iterates, e.g., a distributed system where the gradient is only posessed by the workers and the latter can only be accessed through a stochastic oracle. In general, such existence proof for an $\epsilon$-stationary point is in contradiction to the stochastic gradient and distributed optimization setting, as it may result in an impractical scheme.

As a minor point, it should be noted that the algorithm requires an estimate of $\beta$ and the desired $\epsilon$ as an input to run the algorithm, which are hard to determine-a-priori in practice.

- In the convergence analysis, there are several confusing technical statements which require further clarifications:

-- On page 5, it is stated that $I_t = 1$ implies $|| \nabla f(x_s) || \geq \epsilon$ for $s=1,...,t$. Why is this true? This statement doesn't seem to hold in general.

-- In the proof of Lemma 4, it is not clear what does it mean by "By Markov Inequality, $U_{2\tau} \geq T/2$". To the reviewer's best knowledge, the Markov's inequality bounds the probability of the event that a certain non-negative r.v. is greater than a certain constant. Yet in the statement of Lemma 4 nor in the proof of the lemma, there is no specification of any random event nor its probability. Perhaps the reviewer has missed something from the lemma's statement or from the proof, but at the moment I am unable to deduce the said statement.

-- Also in the proof of Lemma 4, it is stated that $U_{2\tau} \geq T/2$ and $U_{2\tau} \geq T - k(1+2 \tau)$ imply the statement $k \geq T/4(\tau+1)$. Again, the reviewer fails to see why this hold. It seems that the statement would hold instead if $U_{2 \tau} \leq T/2$.

-- The proof of Lemma 3 applies Lemma 2. However, the independence between the stochastic gradients and the iterates should be handled carefully. In particular, Lemma 2 requires $x$ and $N$ to be independent random variables. This may not be the case when the lemma is applied in the proof of Lemma 3. Particularly, the latter involves $x = x_t$ and $N = g_t - \nabla f(x_{t - d_t})$ which may not be independent from each other.

**Time Spent Reviewing:**

5

---

> ### Author Response · Authors · 2021-08-10
> **Thanks for the review - authors response**
>
> Thanks for the very detailed feedback.  Below we address the main concerns in your review.
>
> > “at every update, the worker has to send both $x_{t-d_t}$ and $g_t$ to the server which involves a doubled communication cost”
>
> Note that the communication overhead can be completely eliminated by performing the threshold test **at the worker machine**, reusing $x_t$ to compute $x_{t+1}$ and use that point for the next gradient calculation. See Appendix C in the supplementary for a detailed discussion.
>
> > “it is unclear if the proposed algorithm can be extended to be used with compression”
>
> As we noted above, the thresholding logic can be implemented on the workers, which leaves the communication between the worker and the parameter server (in steps where updates are actually performed) unaffected by the proposed algorithm. Therefore, compression techniques can be implemented independently.
>
> > “it is stated that this success probability can be "amplified" to $1−\delta$ for any $\delta>0$ by repeating the algorithm with $T$ iterations and "running through a point with gradient norm $\leq \epsilon$". Such scheme appears to be impractical since it involves checking the gradient norms of the iterates”
>
> For amplifying the success probability, notice that we do not need to be able to know which of the points is good (i.e. an approximate critical point): all we claim is that by repeating for $\log(1/\delta)$ times, at least one of the points encountered in those repetitions will be good with probability $\geq \delta$. In particular, this **does not** involve computing gradient norms for finding which of the points is good.
>
> We remark that our convergence guarantee (in Theorem 1) is in line with standard guarantees in non-convex optimization, with or without delays.  In practice, for choosing the “right” iterate to output, one often monitors the train/test loss rather than the gradient norm, which can be done with our algorithm as well.
>
> If there are any remaining concerns with this argument, we will be happy to clarify during the discussion.
>
> > "the algorithm requires an estimate of $\beta$ and the desired $\beta$ as an input to run the algorithm, which are hard to determine-a-priori in practice"
>
> Note that for running the algorithm, we do not really need to know $\beta$ nor $\epsilon$ (in the same sense that SGD does not need to know it).  Ultimately there are two meta-parameters that need to be tuned: the step size $\eta$, and the distance threshold (that appears as $\epsilon/2\beta$ in Picky SGD).  Compared to standard algorithms in ML practice (e.g., Adam that has 3 meta-parameters), we think this would require a very reasonable amount of tuning.
>
> ### Technical clarifications:
>
> > “On page 5, it is stated that $I_t=1$ implies $||\nabla f(x_s)|| \geq \epsilon$ for $s=1,...,t$. Why is this true?”
>
> Note that this is w.r.t. the “hypothetical” version of the algorithm (see lines 159--164), that stops updating when $||\nabla f(x_s)|| < \epsilon$. So, $I_t=1$ implies that $||\nabla f(x_s)|| \geq \epsilon$ for all $s=1,\ldots,t$, because otherwise one of these gradient norms would be $< \epsilon$ and the algorithm would have stopped updating, and then it would also hold that $||\nabla f(x_t)|| < \epsilon$, that contradicts the definition of $I_t$.  We will add more explanation around this point---thanks for pointing out!
>
> > “In the proof of Lemma 4, it is not clear what does it mean by "By Markov Inequality, $U_{2\tau} \geq T/2$"”
>
> First off, we remark that the direction of the inequality in line 188 is an unfortunate typo and should be reversed.  By saying “Markov inequality” we meant a **combinatorial Markov argument** rather than a probabilistic one (we realize now this is a confusing way to phrase the argument, and will revise accordingly).  That is, the argument is as follows: The (deterministic!) quantity $U_{2\tau}$ cannot be larger than $T/2$ since this would mean (by definition) that there are more than $T/2$ steps with delay $\geq 2\tau$, which would in turn imply that the average delay is strictly greater than $\tau$, a contradiction.
> If there are any remaining concerns, we will be happy to clarify those during the discussion.
>
> > “in the proof of Lemma 4, it is stated that $U_{2\tau} \geq T/2$ and $U_{2\tau} \geq T−k(1+2\tau)$ imply the statement $k \geq T/4(\tau+1)$. Again, the reviewer fails to see why this hold. It seems that the statement would hold instead if $U_{2\tau} \leq T/2$”
>
> This is a typo indeed. Thanks for noticing!
>
> > “The proof of Lemma 3 applies Lemma 2. However, the independence between the stochastic gradients and the iterates should be handled carefully”
>
> Thank you for spotting this inaccuracy on our part (we will improve in the final version): in Lemma 2, we should have assumed that conditioned on $x$ and $x’$, the noise $N$ is zero-mean and with variance $\leq \sigma^2$.  This of course holds when applying this lemma to prove Lemma 3 (will explain more in the final version): the noise term $N = g_t - \nabla f(x_{t-d_t})$ is indeed zero-mean and with variance $\leq \sigma^2$ when conditioned on the iterates $x_{t-d_t}$ and $x_t$, since $g_t$ was taken at $x_{t-d_t}$ and was received only at round $t$.

---

> > ### Comment · Reviewer_v3Nq · 2021-08-17
> > **.**
> >
> > Thanks for the detailed answer, which addressed some of my concerns especially with the proof of Lemma 2 and Lemma 4. I believe that these are critical issues that should be fixed, especially for getting a clean statement clarifying about the independence between the r.v.s such as $x, x', N$ in Lemma 2.
> >
> > However, I am still not convinced about the success probability amplification argument following Theorem 1. While repeating the algorithm for $\log(1/\delta)$ times indeed "does not involve computing gradient norms for finding which of the points is good", selecting a good point out of the $T \log(1/\delta)$ options would require computing the gradient norms. Though as the response said, the gradient norm is typically replaced by training/testing loss which will be constantly monitored, the overall scheme still involves restarting Picky SGD for multi times. The numerical experiments presented do not seem to adapt such a procedure as the picky SGD is executed only for once. I believe that it might be possible to tighten the analysis and make a more practical statement.
> >
> > In comparison, the analysis with respect to a random stopping scheme, e.g., as done in [Theorem 16, Stich & Karimireddy, 2020], would make more sense for the convergence for stochastic optimization algorithms. Is it possible (or impossible) to establish a bound on $\mathbb{E}[ \| \nabla f(x_t) \|^2 ]$ for picky SGD? It shall be clarified in the paper.
> >
> > P.S. As a minor comment, the "combinatorial Markov argument" in the above sounds like a standard combinatorial proof.

---

> > > ### Author Response · Authors · 2021-08-17
> > > **Thanks for the follow-up**
> > >
> > > Thank you for following up on our response.
> > >
> > > - **Clarifications in proofs**:  Indeed, these are inaccuracies that should and will be (in fact, have already been) fixed.  Thanks again for your outstanding scrutiny.
> > >
> > > - **Amplification argument and restarts**:  It is true that the algorithm we report on in our experiments does not involve restarts---in practice, it turned out that the first repetition already performs very well (compared to well-tuned SGD) and additional repetitions do not improve performance significantly.  It is also true that our theoretical analysis is currently unable to explain this consistent empirical performance of the first repetition.  We will remark about this in the final version.
> > >
> > > - **Random stopping scheme**:  (Essentially repeating our earlier response to iRiS)  There is indeed a difference between our guarantees and those of Stich & Karimireddy: we claim about the best past iterate, while they can argue about the average (i.e., random) one.  Thanks for highlighting this---we will want to remark about it in the final version.  We do not currently know how to analyze random stopping with Picky SGD.  That said, in our view, this difference does not represent a very strong practical limitation: first, even if choosing a random iterate did indeed yield the claimed bound on the expected gradient norm, this would not be very appealing in practice as the variance could be high (put differently, the probability that a good iterate will be picked could be rather low).  Second, in practice---as you mentioned---one very often monitors the train/test loss for choosing which iterate to output, regardless of the gradient norms.  So, in a sense, the typical guarantees we have in non-convex optimization regarding the average gradient norm are really practically useful only for narrowing down the search to $T$ points for which the loss should be evaluated.
> > >
> > > - **Combinatorial Markov argument**: You are correct, this is an entirely standard combinatorial argument and only our phrasing was perhaps somewhat confusing.  We will make this clearer in the final version--thanks again for pointing out.

---

### Official Review · Reviewer_Y4g4 · 2021-07-12

**Rating:** 7
**Confidence:** 4

**Summary:**

This paper considers the problem of distributed stochastic asynchronous optimization, in which workers send delayed stochastic gradients to a server. The focus is on non-convex smooth objectives, although results for convex objectives are provided in the supplementary material.

Standard asynchronous distributed SGD algorithms are first discussed, and the authors highlight that their step-size (and thus, also their iteration complexity) depends on a bound over the maximum possible delay over all iterations. Instead, this paper introduces an algorithm that does not require to know this bound, and only depends on the average observed delay.

This algorithm is called Picky-SGD, and its main difference with standard delayed SGD is that the update is only performed if the delayed iterate is close enough to the current iterate. Theoretical results are then given to show that Picky-SGD achieves the standard complexity results for these problems, but replacing the maximum delay by the average one. The proof is then presented in details, and a set of experiments is given to illustrate the superiority of Picky-SGD over delayed SGD.


**Limitations And Societal Impact:**

No foreseeable societal impact

**Main Review:**

The introduction is very well written, quite thorough, and clear. The contributions (both the algorithm and its analysis) are interesting contributions to this field, and could be leveraged beyond this setting.

The global approach is very intuitive but also quite efficient. Standards algorithms assume a bound on the maximum delay $\tau$, and they reduce the step-size accordingly so that $x_{t - \tau_{\max}}$ is never too far from the current iterate $x_t$ even in the worst case, so that the gradient at $x_{t - \tau_{\max}}$ remains relevant. On the other hand, Picky-SGD directly verifies a condition on $\||x_{t - \tau} - x_t\||$ at each step, and thus avoids having to lower the step-size too much.

Although this seems conceptually simple, work is then needed to show that the algorithm does not discard too many gradients and converges fast enough.

In Picky-SGD, the knowledge of $\tau_\max$ is replaced by the knowledge of a target accuracy for the algorithm. Yet, this is still a hyperparameter to tune, and generally an extra one compared to standard delayed SGD since the bound on the delay only appeared in the step-size, which is generally tuned anyway. I believe that the authors should be significantly more straightforward about this limitation, since it is not so clearly stated anywhere.

Similarly, it is expected that the Picky SGD algorithm outperforms standard delayed SGD since it recovers it when taking $A \rightarrow \infty$ (with the notations of Appendix C.2.2), which corresponds to accepting all iterations. It is good to show that introducing this extra degree of freedom via the acceptance threshold actually helps, but it seems to be at the cost of extra tuning. Again, I believe that this paper would benefit from highlighting this more clearly.

I would be interested in the practical performances of Picky-SGD in the convex / strongly convex cases, in which the theoretical parameters can generally be used (more than in the non-convex case at least), and so in which SGD and Picky-SGD could be compared in a (almost) tuning-free setting. I do not specifically ask for this kind of experiments in the rebuttal though.

Other comments:
I believe that there is a typo in the proof of Lemma 4 (Line 188) in that Markov Inequality should guarantee that $U_{2\tau} \leq T/2$ (and not the other way round). This is consistent with the statement of line 193.

The reference following reference could be relevant:
Hannah, Robert, and Wotao Yin. "On unbounded delays in asynchronous parallel fixed-point algorithms." Journal of Scientific Computing 76.1 (2018): 299-326.

**Time Spent Reviewing:**

2

---

> ### Author Response · Authors · 2021-08-10
> **Thanks for the review - authors response**
>
> Thank you for the valuable comments and the positive feedback!
>
> > “knowledge of $\tau_{\max}$ is replaced by the knowledge of the target accuracy”
>
> Note that in fact we do not need to “know” the target accuracy $\epsilon$: we could have framed our results in terms of the number of steps $T$ rather than $\epsilon$ (this is just a matter of convention), and Theorem 1 would then upper bound the accuracy in terms of $T$ instead of the other way around.  That said, we do indeed need to tune two hyperparameters: a step size and a distance threshold (the $\epsilon/2\beta$ quantity in Picky SGD), which is indeed one additional hyperparameter compared to plain SGD.  We will be more transparent about this in the final version. Thanks!
>
> > "It is good to show that introducing this extra degree of freedom via the acceptance threshold actually helps, but it seems to be at the cost of extra tuning"
>
> The tuning cost for the distance threshold is actually not very significant---we will include details in the revision. Yet again, we definitely agree this should be discussed, and will address this in the final version.
>
> ### Minor comments:
>
> - Typo in Lemma 4:  That is a typo indeed---thanks for spotting!
> - Additional reference:  Thanks for pointing us to this paper - we will cite and discuss in the final version.

---

### Official Review · Reviewer_iRiS · 2021-07-14

**Rating:** 6
**Confidence:** 3

**Summary:**

This paper gives an alternative to SGD, called picky SGD, for stochastic optimization with delays. While Stich & Karimireddy proved that SGD was robust to delays as long as the maximum delay is controled, here picky SGD is proved to be robust to much larger delays, as it requires only the average delay (and not the maximum delay) to be controled. This is supported by both theoretical convergence to a stationary point (in the non-convex case) or to a minimum (in the convex case) and with simulations with artificially introduced delays.

**Limitations And Societal Impact:**

The articulation with the result of Stich & Karimireddy is not detailed enough. I gave suggestions for improving this in the main review.

**Main Review:**

The authors expose clearly their results. The simulations show an important improvement of picky SGD over SGD. However, the code is not provided, thus reproducing the experiments would be quite tedious.

As the central objective of the paper is to replace the dependence in the maximum delay by the average delay, it is natural to ask what empirical evidence do we have that in practical distributed systems, the distribution of delays is heavy tailed. Do you have experiments or references supporting this? The simulations are for heavy-tailed delays that are introduced artificially, thus we could wonder if such delays occur in practice.

The paper presents their result as a variation over a result of Stich & Karimireddy. However, this papers studies more generally non-uniformly bounded noises (their assumption 3). Stich & Karimireddy explain at great length why this is more relevant to machine learning practice. However, the authors of these papers restrict themselves to uniformly bounded noise, with no comment on this restriction. Could you elaborate?
Also, while the theoretical result is well compared with the one of Stich & Karimireddy, the difference in the proof technique is not commented. Is it a variation over the same proof technique? Does picky SGD require completely different techniques?

Finally, I did not succeed in understanding the proof of Theorem 1. It might be only miskates on my side, but I think I need a clarification from the authors.
Mostly, I do not understand the paragraph of l. 159-164.
- What does it mean for the algorithm to "fail"? Does it happen when the algorithm has not found a critical point in a given number of iterations? What is this number of iterations?
- I do not understand the statement "the guarantee of Theorem 1 is valid for this variant if and only if it is valid for the original algorithm". Is it proved somewhere?

In my attempt to understand the definition of "fail", I read the proof of Lemma 4 (I thought that I could understand the statement from the proof). In this proof, two lower bounds are derived for U_{2\tau}. How can you conclude on an inequality between those two bounds?
Also, could you detail the application of Markov inequality? As the delays are arbitrary (and not random), U_{2\tau} is deterministic and I do not understand how Markov inequality intervenes here.


Minor comments:
- l.70: was it necessary to know \tau_max in the work of Stich and Karimireddy?
- why do you assume the function to be non-negative?
- l.151-155: when amplifying, how do you know which of the final points is successful? Gradients are stochastic thus is it not obvious to test if a point is an approximate critical point.


-----------
I increased my grade after the rebuttal, see below an updated review.

**Time Spent Reviewing:**

5

---

> ### Author Response · Authors · 2021-08-10
> **Thanks for the review - authors response**
>
> Thank you for the feedback and the detailed comments.  Below we address the main points you have raised.
>
> > “the code is not provided, thus reproducing the experiments would be quite tedious”
>
> We intend of course to make our code public, but only after publication since it requires clearing some IP-related publishing procedures.  Our experiments will be precisely reproducible as they employ a simulator for the delayed gradients environment.
>
> > “what evidence do we have that in practical distributed systems, the distribution of delays is heavy tailed”;  “The simulations are for heavy-tailed delays that are introduced artificially”
>
> We aimed at fully reproducible experiments, agnostic to specific network/hardware characteristics. Therefore we opted for a simulated statistical delays/schedules approach, allowing full control over the distribution of the delays and complete reproducibility. Our experiments include a range of delay characteristics settings, aimed to capture natural varieties that we have observed in actual real-world training pipelines. One common scenario we encountered is when different workers have very different computing power (at ratios that can reach up to 1:80 !), thereby inducing a heavy-tailed delay distribution.
>
> > “the authors restrict themselves to uniformly bounded noise, with no comment on this restriction. Could you elaborate?"
>
> Our analysis can be extended in a straightforward way to the extended noise model, as is often the case in smooth stochastic optimization (and as done in Stich & Karimireddy [1]).  We chose to focus on the simpler uniform noise model since it is far more common in the literature and since it serves to simplify the analysis (and the resulting bounds) by a bit.  We would be happy to hear in your final comments if you think the paper would become stronger if it addressed the extended model, in which case we will be happy to include such an analysis in our final version.
>
> > “while the theoretical result is well compared with the one of Stich & Karimireddy, the difference in the proof technique is not commented”
>
> Our proof is quite different from that in Stich & Karimireddy---we will elaborate on that more in the final version.  One crucial difference is that Picky SGD selects gradients in an **adaptive manner**, that is, whether a gradient is used or discarded crucially depends (probabilistically) on past iterations (and the randomness therein).  This is represented in our analysis by the indicators $I_t$, and most of the proof is concerned with a careful probabilistic analysis of the “expected movement” of the algorithm (between iterations $t-d_t$ and $t$), which becomes somewhat involved since the $I_t$ are **not** independent of each other.
>
>
> ### Further specific replies:
>
> > “I do not understand the statement "the guarantee of Theorem 1 is valid for this variant if and only if it is valid for the original algorithm"”
>
> This paragraph (lines 159-164) describes an hypothetical algorithm which is **precisely equivalent** to Picky SGD up to the point where they encounter an iterate with gradient norm $\leq \epsilon$, after which point they stop being equivalent.  Note that for the purpose of proving Theorem 1 it suffices to analyze the hypothetical algorithm (which we do): the hypothetical algorithm finds an iterate with gradient norm $\leq \epsilon$ if and only if Picky SGD finds such an iterate!
>
> > “What does it mean for the algorithm to "fail"? Does it happen when the algorithm has not found a critical point in a given number of iterations? What is this number of iterations?”
>
> By “fail” we mean that the algorithm did not go through a point with gradient norm $\leq \epsilon$ within $T$ iterations.  (We term “fail” in line 160---we will make this more explicit in the final version.)  Note that our main theorem says how large $T$ needs to be to ensure that “failure” happens with probability at most $1/2$.
>
> > “In the proof of Lemma 4, two lower bounds are derived for $U_{2\tau}$. How can you conclude on an inequality between those two bounds? Also, could you detail the application of Markov inequality?”
>
> First, the direction of the inequality in line 188 is an unfortunate typo and should be reversed. Once this is fixed, the two bounds can be combined to obtain the conclusion.
> Second, by saying “Markov inequality” we meant a **combinatorial Markov argument** rather than a probabilistic one (we realize now this is a confusing way to phrase the argument, and will revise accordingly).  That is, the argument is as follows: The (deterministic!) quantity $U_{2\tau}$ cannot be larger than $T/2$ since this would mean (by definition) that there are more than $T/2$ steps with delay $\geq 2\tau$, which would in turn imply that the average delay is strictly greater than $\tau$, a contradiction.
> If there are any remaining concerns, we will be happy to clarify those during the discussion.
>
> ### Further replies to minor comments:
>
> > “l.70: was it necessary to know $\tau_{\max}$ in the work of Stich and Karimireddy?”
>
> Yes - it affects the optimal choice of the step size parameter (that yields the convergence bound we compare to).
>
> > “why do you assume the function to be non-negative?”
>
> This is without loss of generality when considering gradient methods - one can simply shift the function to have minimal value zero without affecting its gradients.  In other words, we could have equivalently assumed that $f(x_1)-f(x^*) \leq F$ without assuming that $f(x^*) \geq 0$.
>
> > “151-155: when amplifying, how do you know which of the final points is successful? Gradients are stochastic thus is it not obvious to test if a point is an approximate critical point”
>
> Note that Theorem 1 only guarantees that one of the points $x_1,\ldots,x_T$ is good (i.e. an approximate critical point).  This is in line with standard convergence guarantees for non-convex optimization, with or without delays.  For amplifying the success probability, notice that we do not need to be able to know which of the points is good: all we claim is that by repeating for $\log(1/\delta)$ times, at least one of the points encountered in those repetitions will be good with probability $\geq \delta$.
>
> Again, if there are any remaining concerns with this argument, we will be happy to clarify during the discussion.

---

> > ### Comment · Reviewer_iRiS · 2021-08-12
> > **updated review after the author response**
> >
> > I thank the authors for their detailed answers. Thanks to their clarifications, I was able to go much further in the understanding of the proof technique, that I now believe to be rigorous. I have thus increased my grade. I still have some remaining concerns:
> >
> > - In their author response, the authors motivate to consider heavy-tailed delays only from a personal experience. These experiments on real-world systems are not reported in the paper. As this is the central motivation for the paper, we could expect to have stronger arguments that this is relevant.
> >
> > - In their author reponse, the authors claim that their paper can be generalized to more sophisticated noise models easily. I do think this would strengthen the paper (again, the motivations are detailed by Stich & Karimireddy). However, I am quite skeptical that it would be that easy. Moreover, there are two possible generalizations, corresponding to Assumptions 3 and 3* of Stich & Karimireddy. What is your precise claim? That the non-convex case can be generalized to Assumption 3 and the convex case of Assumption 3*?
> >
> > - Finally, to the best of my knowledge, an important difference with Stich & Karimireddy is not commented: from a quick look at their Theorem 16, it seems that the performance guarantees of Stich & Karimireddy apply to a randomly sampled past iterate, while the authors prove a result for the best of the past iterates, which is impossible to determine in practice. This is a strong practical limitation, although from the simulations it seems that the last iterate of Picky SGD performs well. Am I misunderstanding something? Could the result of the authors be generalized to a randomly sampled past iterate?

---

> > > ### Author Response · Authors · 2021-08-15
> > > **Thanks for the prompt update**
> > >
> > > Thank you for the attention to details and the prompt reassessment.
> > >
> > > - **Empirical motivation for heavy-tailed delays:**  We agree: sharing the real-world empirical evidence/experiments would strengthen the motivation for our result.  Unfortunately, delays are a byproduct of the particular computing infrastructure, the details of which are proprietary information we are unable to disclose.  Nevertheless, we believe that the scenarios of heavy-tailed delays we allude to in the paper are very natural, and the academic interest in analyzing the general heavy-tailed setting sufficiently motivates our results.
> > >
> > > - **Generalized noise model:** Thank you for confirming interest in such an extension---in light of your input, we will work out the details for the final version.  In our response above, we were referring to an analysis for the non-convex case (which is the main focus of our paper), for which Assumption 3 from Stich & Karimireddy is relevant.  We still believe that such an extension should not be very hard to establish, but let us update after we engage in this.
> > >
> > > - **Best vs. average past iterates:** There is indeed a difference between our guarantees and those of Stich & Karimireddy: we claim about the best past iterate, while they can argue about the average (i.e., random) one.  Thanks for highlighting this---we will want to remark about it in the final version.  That said, in our view, this difference does not represent a very strong practical limitation: first, while choosing a random iterate does indeed yield the claimed bound on the expected gradient norm, this would not be very appealing in practice as the variance could be high (put differently, the probability that a good iterate will be picked could be rather low).  Second, in practice one very often monitors the train/test loss for choosing which iterate to output, regardless of the gradient norms.  So, in a sense, the typical guarantees we have in non-convex optimization regarding the average gradient norm are really practically useful only for narrowing down the search to $T$ points for which the loss should be evaluated.

---

> > > > ### Comment · Reviewer_iRiS · 2021-08-15
> > > > **Thank you**
> > > >
> > > > Thank you for your clarifications.

---

### Official Review · Reviewer_TNHX · 2021-07-26

**Rating:** 7
**Confidence:** 4

**Summary:**

This paper improves the iteration complexity to get $\epsilon$ small gradients for smooth non-convex functions using SGD with delayed gradients. This setting, in particular, captures asynchronous distributed SGD. The previously known best algorithm [1] accumulates error feedback to provide a $\sigma^2/\epsilon^4 + \tau_{max}/\epsilon^2$ rate, where $\tau_{max}$ is the maximum delay in any update. This can be catastrophic in cross-device federated learning with edge devices, where a device gets disconnected for multiple rounds, and then sends its gradient for the update.

This paper deals with such a scenario in a very simple manner, by measuring the distance between the current iterate and the old iterate at which the gradient was computed, and making the update only if it is smaller than a threshold value. This algorithm is aptly called *picky SGD*. Surprisingly, this removes the need for error feedback and is able to guarantee a $\sigma^2/\epsilon^4 + \tau_{avg}/\epsilon^2$ rate where $\tau_{avg}$ is the average delay over all the rounds. Picky SGD with a different threshold also offers state-of-the-art convergence rates for convex smooth functions, with the dependence on the maximum delay changed to the one on average delay.

The authors also study the effect of sampling the delays from distributions with different levels of variance in their experiments. The results demonstrate that picky SGD requires comparable or fewer steps to reach a fixed level of accuracy than vanilla asynchronous SGD. This is attributed to the fact that picky SGD tolerates a much larger step size than asynchronous SGD.



**References**

[1] Stich, Sebastian U., and Sai Praneeth Karimireddy. "The error-feedback framework: Better rates for SGD with delayed gradients and compressed communication." arXiv preprint arXiv:1909.05350 (2019).

**Limitations And Societal Impact:**

Some experimental limitations as mentioned in the main review.

**Main Review:**

This is a well-written paper, with a clear description of the goals, contributions, results, and proof sketch. The proposed algorithm is in fact simpler (albeit at the cost of storing the past iterates) than the error-corrected SGD but still improves upon its convergence guarantee, making the upper bound *data-dependent*. I have some concerns though, and I would be happy to increase my score if they are resolved.

1) Can the analysis be extended to a more general noise model like the strong growth model as in [1]?

2) In the theoretical guarantees the paper compares to error-correcting SGD. However, why was it not included as a baseline in the experiments?

3) It is useful to see the magnified tail of the plot, but it might make more sense to plot the y-axis on a log scale. Further, the experiments should be repeated multiple times to kill some noise in the curves and understand the variance of these algorithms. In fact, it would be good to include repetitions for the same delay schedule as well as different delay schedules sampled from the same distribution.

4) It is not clear how the algorithms stop making updates in the experiments. Were they stopped after reaching a particular target accuracy? It doesn't seem to be the case looking at the figure. In setting (D) SGD stops earlier and vice versa for setting (C).

5) While picky SGD is simpler to analyze and discuss in theory, it has two limitations in practice. First tuning the for the threshold $\epsilon/2\beta$, and second storing the past iterates only up to some point in history. How to go around these issues in practice? It is important to show how robust the algorithm is to each of these choices. Also, since memory costs can often be the bottleneck, some wall clock time experiments would also be useful, again over some reasonable range of the tunable parameters. These limitations can make EC-SGD [1] a better choice in some settings because we need to tune it as much as usual SGD.

**Typos**

1) L58 previous art -> previous state of the art

2) L283 SGD -> SGD's

# Final Verdict

I liked this paper overall except for some minor concerns and suggestions. They have been addressed, and the authors have promised to add some important discussions in the revised version. In light of that, I am improving my score.


**References**

[1] Stich, Sebastian U., and Sai Praneeth Karimireddy. "The error-feedback framework: Better rates for SGD with delayed gradients and compressed communication." arXiv preprint arXiv:1909.05350 (2019).



**Time Spent Reviewing:**

8

---

> ### Author Response · Authors · 2021-08-10
> **Thanks for the review - authors response**
>
> Thank you for the feedback and the thoughtful comments.  Below we address the main points you have raised.
>
> > “Can the analysis be extended to a more general noise model like the strong growth model as in [1]?”
>
> Yes - indeed the analysis can be extended in a straightforward way to the extended noise model, as is often the case in smooth stochastic optimization (and as done in Stich & Karimireddy [1]).  We chose to focus on the simpler uniform noise model since it is far more common in the literature and since it serves to simplify the analysis (and the resulting bounds) by a bit.  We would be happy to hear in your final comments if you think the paper would become stronger had it addressed the extended model, in which case we will be happy to include such an analysis in our final version.
>
> > “In the theoretical guarantees the paper compares to error-correcting SGD. However, why was it not included as a baseline in the experiments?”
>
> Note that EC-SGD is an abstract meta-algorithm proposed in Stich & Karimireddy [1] to analyze several different optimization algorithms in a unified manner.  For the case of delayed gradients, which is the focus of our paper, EC-SGD simply becomes SGD with step size smaller by a factor of $\tau_{\max}$---see the algorithm called D-SGD in [1].  So, in fact, we did compare with this algorithm (SGD) in the experiments, searching over a wide range of step sizes.  We will make sure this becomes clear in the final version.  Thanks!
>
> > “it might make more sense to plot the y-axis on a log scale”;  “experiments should be repeated multiple times to kill some noise in the curves”;  “it would be good to include repetitions for the same delay schedule as well as different delay schedules sampled from the same distribution”
>
> Since repetitions are very costly, we validate the superiority of our scheme by having consistent results across different delay schedules and learning rate regimes. Moreover, the delay schedule realization was kept the same (across the compared optimization algorithms) for comparison purposes, but nevertheless, our (not reported) experiments on various realizations result in the same qualitative superiority. As for the log-scale suggestion, if differences are better illustrated that way (we will generate plots and check) then we will definitely replace the y-axis scale in the figures as suggested.
>
> > “It is not clear how the algorithms stop making updates in the experiments”
>
> Thanks for spotting---our intention was to run each experiment up to reaching a prespecified accuracy; all of the reported experiments ran beyond this point, some more than others. In any case, we will fix and recreate all plots using the exact same number of epochs (700) for all runs.
>
> > “While picky SGD is simpler to analyze and discuss in theory, it has two limitations in practice. First tuning the for the threshold $\epsilon/2\beta$, and second storing the past iterates only up to some point in history”
>
> The communication overhead can be completely eliminated by performing the threshold test at the worker side, reusing $x_t$ to compute $x_{t+1}$ and use that point for the next gradient calculation. See Appendix C in the supplementary for a detailed discussion. Regarding $\epsilon/2\beta$, in the experimental section we treat it as a meta-parameter and perform a standard search, however, in practice we found it very easy to tune by simply logging the distances between $x_t$ and $x_{t-d_t}$ and choosing a value that is slightly above the typical distance value. We will add a note explaining this approach in the final version.
>
> > “Since memory costs can often be the bottleneck, some wall clock time experiments would also be useful”
>
> As we noted above, the overhead can be completely eliminated by performing the checks in the workers, as detailed in Appendix C. Since the memory overhead amounts to storing an additional copy of the parameters in the workers we do not expect a significant effect on wall time.

---

> > ### Comment · Reviewer_TNHX · 2021-08-17
> > **Response**
> >
> > > We would be happy to hear in your final comments if you think the paper would become stronger had it addressed the extended model, in which case we will be happy to include such an analysis in our final version.
> >
> > It would be good to also have this, though admittedly it doesn't make the paper much stronger itself.
> >
> > > So, in fact, we did compare with this algorithm (SGD) in the experiments, searching over a wide range of step sizes. We will make sure this becomes clear in the final version.
> >
> > Ok.
> >
> > > We will add a note explaining this approach in the final version.
> >
> > That would be helpful for anyone trying to implement and tune this in the future. Thanks!

---

### Decision · Program_Chairs · 2021-09-27

**Decision:**

Accept (Poster)

**Comment:**

This paper studies the convergence of gradient descent with delayed updates. While most prior results depended on a parameter corresponding to the largest encountered delay, this paper derives convergence analysis depending on just the average delay.
The presented algorithm does not need to know the average delay parameter beforehand.

The reviewers found this an overall solid technical contribution and recommend acceptance.

The authors are strongly encouraged to take the reviewer's feedback into account when preparing the revision, as a few concerns remain:
- reviewer v3Nq spotted some inaccuracies in the proofs, which the author's promised to fix
- while indeed prior methods need 'to know' the maximum delay to set the theoretical step size, in practice the tuning is limited to a single parameter only (the stepsize). The proposed scheme has two hyperparameters (stepsize and threshold), and the reviewers concerns on practical benefits seem justified. It would be great if these comments could be addressed diligently in the revision.

Additionally, I would encourage the authors to comment on the concurrent work [[Aviv et al, Learning Under Delayed Feedback: Implicitly Adapting to Gradient Delays, ICML 2021](http://proceedings.mlr.press/v139/aviv21a/aviv21a.pdf)] published at ICML earlier this year. It would be great to explain the key differences and similarities of the proof techniques and results to the readers.